# Exposure factors associated with SARS-CoV-2 seroprevalence during the first eight months of the COVID-19 pandemic in the Netherlands: A cross-sectional study

Demi M. E. Pagen[1,2]*, Stephanie Brinkhues[1,2], Nicole H. T. M. Dukers-Muijrers[1,3], Casper D. J. den Heijer[1,2], Noortje Bouwmeester-Vincken[4], Daniëlle A. T. Hanssen[5], Inge H. M. van Loo[5], Paul H. M. Savelkoul[5,6], Christian J. P. A. Hoebe[1,2,5]

1 Department of Sexual Health, Infectious Diseases and Environmental Health, South Limburg Public Health Service, Heerlen, The Netherlands, 2 Department of Social Medicine, Care and Public Health Research Institute (CAPHRI), Maastricht University, Maastricht, The Netherlands, 3 Department of Health Promotion, Care and Public Health Research Institute (CAPHRI), Maastricht University, Maastricht, The Netherlands, 4 Department of Infectious Diseases, North Limburg Public Health Service, Venlo, The Netherlands, 5 Department of Medical Microbiology, Maastricht University Medical Centre (MUMC+), Care and Public Health Research Institute (CAPHRI), Maastricht University, Maastricht, The Netherlands, 6 Department of Medical Microbiology and Infection Control, Amsterdam Medical Centre, Amsterdam, The Netherlands

* Demi.Pagen@ggdzl.nl

**Data Availability Statement:** Data cannot be shared publicly because the data contains

## Abstract

### Background

The availability of valid Severe Acute Respiratory Syndrome Coronvirus-2 (SARS-CoV-2) serological tests overcome the problem of underestimated cumulative Coronavirus Disease 2019 (COVID-19) cases during the first months of the pandemic in The Netherlands. The possibility to reliably determine the number of truly infected persons, enabled us to study initial drivers for exposure risk in the absence of routine testing. Numerous activities or circumstances can accelerate virus spread, here defined as exposure factors. Hence, we aimed to evaluate a wide variety of demographic, behavioural and social exposure factors associated with seropositivity during the first eight months of the pandemic in Limburg, The Netherlands.

### Methods

SARS-CoV-2 point-seroprevalence was determined cross-sectionally to indicate previous infection in a convenience sample of minimal 10,000 inhabitants of the study province. All adult (18+ years) inhabitants of the study province were eligible to register themselves for participation. Once the initial 10,000 registrations were reached, a reserve list was kept to ensure sufficient participants. Possible exposure factors were mapped by means of an extensive questionnaire. Associated exposure factors were determined using univariable and multivariable logistic regression models.

### Results

Seropositivity was established in 19.5% (n = 1,948) of the 10,001 participants (on average 49 years old (SD = 15; range 18–90 years), majority women (n = 5,829; 58.3%). Exposure

potentially identifying and sensitive participant information. Due to the General Data Protection Regulation, it is not allowed to distribute or share any personal data that can – directly or indirectly – be traced back to an individual. Besides, publicly sharing the data would not be in accordance with participants' consent obtained for this study. Therefore, data are available from the head of the data-archiving of the Public Health Service South Limburg on reasonable request (Helen Sijstermans: helen.sijstermans@ggdzl.nl).

**Funding:** This work was financially supported by the Province of Limburg, the Netherlands. The funders had no role in study design, data collection and analysis, decision to publish, or preparation of the manuscript.

**Competing interests:** The authors have declared that no competing interests exist.

factors associated with seropositivity included current education, working in healthcare and not working from home, and being a member of three or four associations or clubs. Specifically for February-March 2020, visiting an après-ski bar during winter sports in Austria, travelling to Spain, celebrating carnival, and participating in a singing activity or ball sport were associated with seropositivity.

## Conclusions

Our results confirm that relevant COVID-19 exposure factors generally reflected circumstances where social distancing was impossible, and the number and duration of contacts was high, in particular for indoor activities.

## Introduction

By November 2021, there have been more than 246 million confirmed coronavirus disease 2019 (COVID-19) cases worldwide since the start of the pandemic, accompanied by approximately 5 million COVID-related reported deaths [1]. On 27[th] February 2020, the first COVID-19 case was confirmed in the Netherlands [2]. Initially, the province of Limburg was one of the most heavily affected Dutch provinces, with 439 confirmed cases per 100,000 inhabitants until July 2020 and the highest hospitalization and mortality rate nationally [3]. Average mortality was exceeded with 62% for this province during the first nine weeks of the pandemic [4]. Burden continued during the second wave of infections, when the southern provinces remained the most affected areas regarding COVID-related deaths as well [3]. Over time, new virus variants emerged, maintaining new waves of infections. These new variants display alternations in transmissibility and infectivity [5], and thereby continuously put pressure on defining the most effective infection prevention measures to combat the exponential growth of new COVID-19 cases.

In the Netherlands, the total number of COVID-19 cases was largely underestimated during the first months of the pandemic. Patients were initially only tested for COVID-19 when clinical symptoms were present (i.e. fever ($\geq$ 38˚C) and dyspnoea and/or cough) together with an epidemiological link (i.e. contact with a confirmed case or travel to an high COVID-19 incidence area) [6]. This underestimation limited the possibility to assess which factors contributed to the primary spread of severe acute respiratory syndrome coronavirus 2 (SARS-CoV-2).

Numerous activities or circumstances can accelerate virus spread, here defined as exposure factors. Since SARS-CoV-2 can be transmitted directly and by air, social distancing with a minimum of one to two meters is assumed to be sufficient to avoid spreading [7]. Furthermore, air quality and adequate ventilation is key, as shown in an experimental setting where SARS-CoV-2 remained viable in aerosols for the duration of three hours [8]. The application of social distancing and ventilation measures are unequally spread over settings and contexts, such as types of occupation. A significantly increased risk of COVID-19 infection was established among frontline healthcare workers in the United Kingdom [9]. A Norwegian population-based study showed that other healthcare workers (e.g. nurses, physicians, dentists and physiotherapists), people working in hospitality industry and in (public) transport also had a higher odds of a COVID-19 infection confirmed by positive polymerase chain reaction (PCR) during the first two waves of infection [10]. Just before infection prevention measures were implemented in March 2020, many mass events such as the annual carnival celebrations took

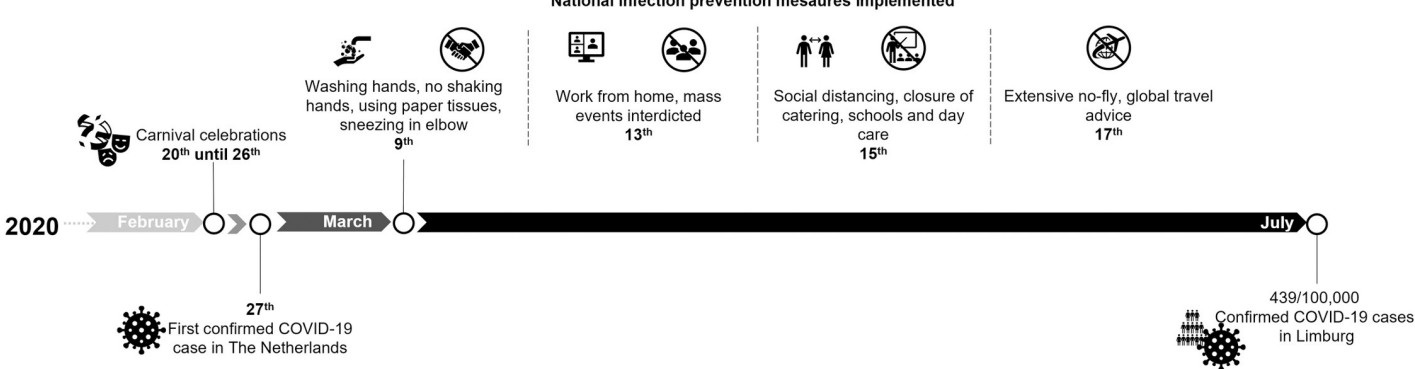

**Fig 1. Timeline depicting events, first COVID-19 case, implementation of prevention measures and cumulative confirmed COVID-19 cases.**

place (Fig 1). Subsequent interdiction of mass events by the implemented measures, challenged the opportunity to measure their contribution to virus spreading at later times.

To better understand initial drivers for exposure risk in the absence of routine testing at that time, serology can be used to identify (unnoticed) infections. Defining the first–untested and therefore unconfirmed–COVID-19 cases is essential, as implementation of infection prevention measures largely eliminated exposure factors which were abundant in the first period. Serological testing in an unvaccinated population allows to reliably determine the number of truly infected persons. In 92% of the cases, COVID-19 antibodies are still detectable seven months after infection [11]. Hence, our study aimed to evaluate seroprevalence exposure factors associated with seropositivity during the first eight months of the pandemic, assessed prior to vaccination. Determining these exposure factors that facilitate transmission can be useful to inform future infection prevention policies and potentially confirm the effectiveness of existing measures.

## Method

### Study design

A cross-sectional study was designed to assess exposure factors which contributed to the spread of SARS-CoV-2 in the first eight months of the pandemic. A convenience sample of 10,000 inhabitants of a southern Dutch province was used, reflecting about 1% of 1 million adults in this province. Point-seroprevalence was determined to indicate previous infection. Possible exposure factors were mapped by means of an extensive questionnaire. A convenience sample was chosen to accommodate equal opportunities for the general population in their great demand on to know about their COVID-19 serostatus.

### Participants

All adult inhabitants of 18 years and older of our study province (Limburg) were eligible for participation until sufficient registrations were reached. After the first 10,000 registrations were reached, registrations were automatically allocated to the reserve list. When initial registered participants declined participation, adults registered on the reserve list were invited (first-come-first-serve) to ensure sufficient participants (minimally 10,000). Participants needed to speak, read, and write Dutch language, as the questionnaire was only available in Dutch. This applies to 95% of our study province inhabitants [12].

## Data collection

Data collection was completed between 28[th] October and 23[rd] December 2020 and comprised donating a blood sample and filling out an online questionnaire.

Online participation registration was conducted via market research software from Crowd-tech (ISO-20252 and ISO-27001 certified, London, UK). To register, personal and contact details were required. An appointment for blood drawing was made by phone at one of the four test locations (Maastricht, Urmond, Landgraaf, and Venlo). One 10 ml ethylenediamine-tetraacetic acid (EDTA) tube was taken by venepuncture by trained and certified health professionals. All blood samples were kept at room temperature and transported to the Medical Microbiology Laboratory of the Maastricht University Medical Centre+ (MUMC+). Samples were tested for total COVID-19 antibodies using the Wantai SARS-CoV-2 Ab enzyme-linked immunoassay (ELISA) test (Beijing Wantai Biological Pharmacy Enterprise Co., Ltd., Beijing, China) in accordance with the manufacturer's instructions [13]. This test was chosen after showing best performance in a multicentre evaluation by the National Institute for Health and Environment [14]. Samples were classified by optical density value as being positive, borderline, or negative. In our study, eight samples had borderline values. Based on pilot experience using the Wantai among five hospital employees with serum pairs, four out of five borderline values seroconverted to positive in a second serum sample. Therefore, we decided to classify the borderline values as positives. Participants were informed about their serostatus via email within three weeks after providing the blood sample.

## Exposure measures

The online questionnaire was sent by email at least two days after providing the blood sample. The questionnaire covered: general demographics, current education, occupation and opportunity to work from home, memberships to associations or clubs, and specific behaviours performed during the first months (February-March 2020) of the pandemic.

Demographics included gender, age, place of residence and level of education. Level of education was categorized into practically trained (no, lower general, lower vocational, general secondary, and secondary vocational education) and theoretically trained (higher general, pre-university, higher professional, and scientific education). Based on place of residence, two geographical regions of the study province were established: northern and southern. A variety of occupations was listed, such as healthcare, education, day-care, and catering industry. Working in healthcare was further defined by specific sectors. The degree of working from home was determined for different time periods: February-March, April-May, and June-November 2020. Working from home was dichotomized into not working from home during all periods and (partly) working from home during at least one of the periods. Membership to a variety of associations or clubs was assessed and being member to a music association was further specified. Memberships to multiple associations or clubs with a perceived higher risk of infection were merged, including membership to a sport, youth, volunteer, social or traditional association of the study province and a fanfare, brass or jazz band were summed up. To grasp all singing activities together as an exposure factor we combined being member of a choral society, a (church)choir or participated in a singing activity. Specific behaviours performed in February-March 2020 comprised travelling for winter sports or other purposes, celebrating carnival, and attending specific activities. For travelling, country of destination was specified supplemented with extent to which participants visited an après-ski bar when travelled for winters sport (i.e., not, couple of days, or majority of the days). For all carnival days separately the amount of time spend inside was counted. A sum score was calculated for attending carnival celebrations inside for all carnival days combined. The score was divided in four categories based on

quartiles. Eventually, involvement in 48 different activities where at least 30 people were present was counted. Activities were categorized in religious ceremonies, attractions, events, cultural activities, hobbies, sports, going out, and others. For nine activities the specific date and place of the activity were indicated (attending a wedding or funeral, visiting a museum, convention, charity event, festival, sport event, professional soccer match, or taking a day trip by bus or boat). A telephone helpline was available to facilitate assistance for participants who were unable to fill out the questionnaire themselves.

## Statistical analysis

Only complete participation was taken into analysis, meaning participation with both a blood sample and questionnaire. No missing data had to be handled, since all questions in the questionnaire were mandatory.

Seroprevalence was a dichotomous outcome measure (positive or negative). Seroprevalence was used to identify possible exposure factors, meaning factors where proportion of seropositive participants exceeded the average. Univariable and corrected multivariable logistic regression analysis were used to study the association between possible exposure factors and seropositivity. Exposure factors with a p-value <0.05 in univariable logistic regression were retained in the multivariable model and multicollinearity between exposure factors was checked. Interactions between exposure factors and geographical region were tested. No geographical differences in exposure factors were established. A p-value of <0.05 was considered statistically significant. Data were analysed using Statistical Package for the Social Sciences (SPSS) (version 26.0, IBM, Armonk, USA).

## Ethical statement

The study protocol, participant information form and written informed consent form were reviewed and approved by the Medical Ethical Committee of the MUMC+ (NL74791.068.20/ METC20-071). The study is registered at the Netherlands Trial Register (NL8889).

## Results

In total 10,108 participants provided a blood sample and 10,001 participants additionally completed the questionnaire, which was taken as the study population (Fig 2). Main reasons for declining participation included: changed their mind, did not show during appointment, unable to reach, and current COVID-19 infection.

## Study population versus source population

About 1% of the adult inhabitants of all 42 municipalities in the study province participated: ranging from 0.2% to 1.5% within the various municipalities. The age distribution of participants was comparable to the source population, but the study population included relatively more women; 58.3% versus 50.3% in the source population.

## Sociodemographic characteristics of the study population

Participants were on average 49 years of age (standard deviation (SD): 15, range 18–90) and 52.3% of the participants were theoretically trained. A considerable number of participants worked in healthcare (Table 1). Some participants (n = 286) were aware of their prior COVID-19 infection, PCR confirmed.

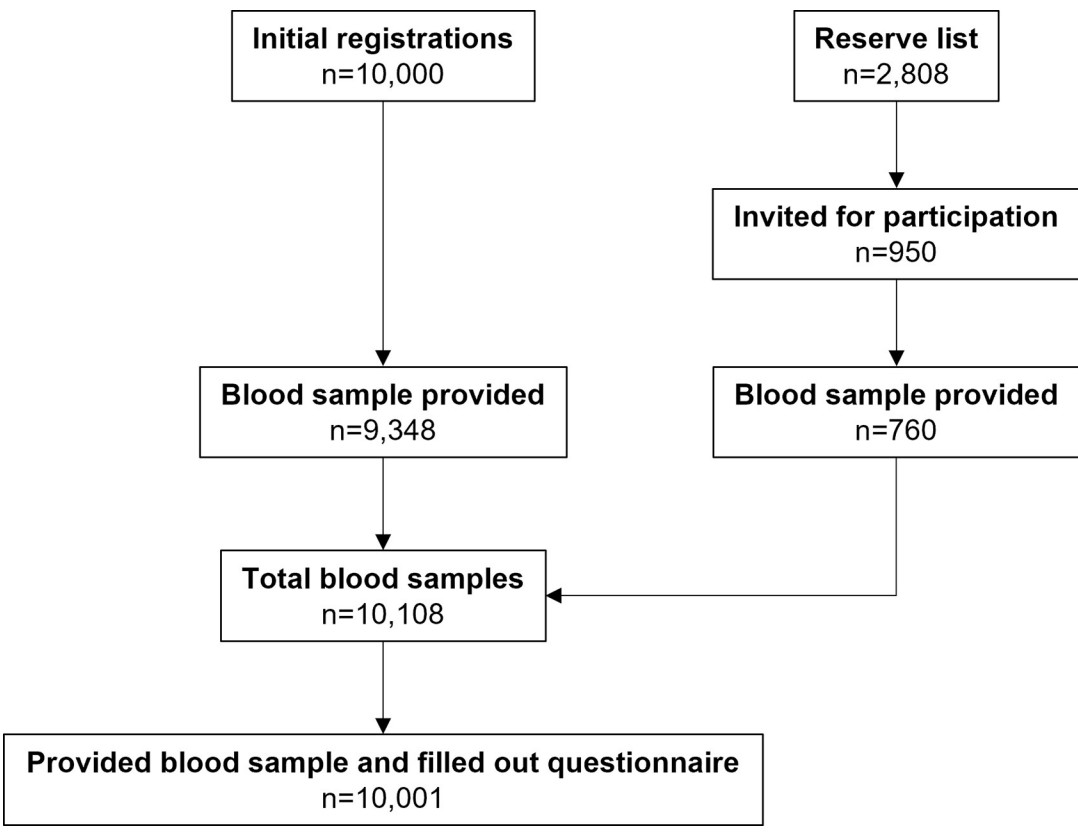

**Fig 2. Flow chart of total study population (n = 10,001) stratified for initial registrations and reserve registrations.**

### Seroprevalence

Overall, 1,948 of the 10,001 participants tested positive for presence of COVID-19 antibodies (19.5% [95% Confidence Interval (CI):18.7%-20.3%]). Seroprevalence varied geographically and was higher in the northern part (23.5% [95% CI:22.2%-24.9%]) than the southern part (17.0% [95% CI:16.0%-17.9%]). A difference in seroprevalence was observed between men (18.7% [95% CI:17.5%-19.9%]) and women (20.7% [95% CI:19.0%-21.0%]). A relatively high seroprevalence was observed among participants aged between 18 and 29 years of age (22.8% [95% CI:20.4%-25.1%]) and practically trained participants (20.6% [95% CI:19.5%-21.8%]) (Table 1).

**Table 1. Characteristics of total study population (n = 10,001) and stratified for serostatus.**

|  |  | Total n = 10,001 |  |  | Seronegative n = 8,053 |  | Seropositive n = 1,948 |  |
|---|---|---|---|---|---|---|---|---|
| **Gender** | **SP (%)** | **95% CI** | **n** | **%** | **n** | **%** | **N** | **%** |
| Male | 18.7 | 17.5–19.9 | 4,167 | 41.7 | 3,389 | 42.1 | 778 | 39.9 |
| Female | 20.0 | 19.0–21.0 | 5,829 | 58.3 | 4,662 | 57.9 | 1,167 | 59.9 |
| Other | - |  | 5 | 0 | 2 | 0 | 3 | 0.2 |
| Age (mean (SD)) | - |  | 49 (15) |  | 49 (15) |  | 50 (15) |  |
| Age |  |  |  |  |  |  |  |  |
| 18–29 years | 22.8 | 20.4–25.1 | 1,208 | 12.1 | 933 | 11.6 | 275 | 14.1 |

(*Continued*)

**Table 1.** (Continued)

| Gender | Total n = 10,001 | | | | Seronegative n = 8,053 | | Seropositive n = 1,948 | |
|---|---|---|---|---|---|---|---|---|
| | SP (%) | 95% CI | n | % | n | % | N | % |
| 30–39 years | 16.5 | 14.7–18.2 | 1,676 | 16.8 | 1,400 | 17.4 | 276 | 14.2 |
| 40–49 years | 16.9 | 15.2–18.7 | 1,766 | 17.7 | 1,467 | 18.2 | 299 | 15.3 |
| 50–59 years | 21.0 | 19.4–22.7 | 2,393 | 23.9 | 1,890 | 23.5 | 503 | 25.8 |
| 60–69 years | 20.3 | 18.6–22.0 | 2,163 | 21.6 | 1,723 | 21.4 | 440 | 20.3 |
| 70–79 years | 19.4 | 16.6–22.3 | 746 | 7.5 | 601 | 7.5 | 145 | 7.4 |
| 80+ years | 20.8 | 9.3–32.3 | 49 | 0.5 | 39 | 0.5 | 10 | 0.5 |
| Geography | | | | | | | | |
| Northern part | 23.6 | 22.7–24.4 | 3,770 | 37.7 | 2,882 | 35.8 | 888 | 45.6 |
| Southern part | 17.0 | 16.3–17.7 | 6,231 | 62.3 | 5,171 | 64.2 | 1,060 | 54.4 |
| Level of education | | | | | | | | |
| Practically trained | 20.6 | 19.5–21.8 | 4,768 | 47.7 | 3,784 | 47.0 | 984 | 50.5 |
| Theoretically trained | 18.4 | 17.4–19.4 | 5,233 | 52.3 | 4,269 | 53.0 | 964 | 49.5 |
| Occupation | | | | | | | | |
| Healthcare | 25.8 | 23.6–27.9 | 1,567 | 15.7 | 1,163 | 14.4 | 404 | 20.7 |
| Catering industry | 23.8 | 18.7–28.9 | 269 | 2.7 | 205 | 2.5 | 64 | 3.3 |
| Non-medical service | 22.1 | 15.5–28.6 | 154 | 1.5 | 120 | 1.5 | 34 | 1.7 |
| Youth care | 18.8 | 12.8–24.7 | 165 | 1.6 | 134 | 1.7 | 31 | 1.6 |
| Education | 18.5 | 15.5–21.5 | 644 | 6.4 | 525 | 6.5 | 119 | 6.1 |
| Day-care | 18.3 | 11.1–25.6 | 109 | 1.1 | 89 | 1.1 | 20 | 1.0 |
| Retail | 18.7 | 14.7–22.6 | 375 | 3.7 | 305 | 3.8 | 70 | 3.6 |
| Other transport and storage | 23.7 | 16.7–30.8 | 139 | 1.4 | 106 | 1.3 | 33 | 1.7 |
| Administration | 20.1 | 17.1–23.0 | 693 | 6.9 | 554 | 6.9 | 139 | 7.1 |
| Knowledge work | 16.3 | 13.8–18.8 | 830 | 8.3 | 695 | 8.6 | 135 | 6.9 |
| Technical | 19.1 | 16.4–21.8 | 801 | 8.0 | 648 | 8.0 | 153 | 7.9 |
| Government | 17.6 | 14.9–20.3 | 766 | 7.7 | 631 | 7.8 | 135 | 6.9 |
| Emergency services[a] | 10.4 | 5.3–15.6 | 134 | 1.3 | 120 | 1.5 | 14 | 0.7 |
| Media and communication | 11.9 | 7.5–16.3 | 210 | 2.1 | 185 | 2.3 | 25 | 1.3 |
| Other[b] | 17.3 | 15.9–18.7 | 318 | 3.2 | 263 | 3.3 | 55 | 2.8 |
| Not working | 18.3 | 14.1–22.5 | 2,827 | 28.3 | 2,310 | 28.7 | 517 | 26.5 |
| Type of healthcare facility | | | | | | | | |
| Disabled care | 37.1 | 29.8–44.4 | 167 | 1.7 | 105 | 1.3 | 62 | 3.2 |
| Home care | 26.6 | 19.3–33.9 | 139 | 1.4 | 102 | 1.3 | 37 | 1.9 |
| Nursing home | 31.1 | 24.0–38.2 | 163 | 1.6 | 112 | 1.4 | 51 | 2.6 |
| Hospital | 26.5 | 20.8–32.2 | 234 | 2.3 | 172 | 2.1 | 62 | 3.2 |
| Other[c] | 22.2 | 19.4–25.0 | 864 | 8.6 | 672 | 8.3 | 192 | 9.8 |
| Not working in healthcare | 18.3 | 17.5–19.1 | 8,434 | 84.3 | 6,890 | 85.6 | 1,544 | 79.3 |

Note.- CI, confidence intervals; SD, standard deviation; SP, seroprevalence.

[a] Excluding ambulance service.

[b] Process industry, agriculture and horticulture, public transport, transport of fuels and waste or other.

[c] Ambulance service, physiotherapy and occupational therapy, intramural and extramural mental healthcare, general practice, youth care, maternity care or obstetric practice, rehabilitation clinic, dental practice, small scale housing, residential care institution or other.

Values are n (%) unless otherwise specified. Seroprevalence represents the proportion of seropositive participants in the total study population. Percentages per category are presented for total study population, seronegative and seropositive populations.

## Exposure factors associated with seropositivity (Table 2)

Participants following vocational secondary education or scientific education had an increased odds for testing seropositive (Odds Ratio (OR) = 1.88 [95% CI:1.10–3.23] and 1.60 [95% CI:1.14–2.25], respectively).

**Table 2. Exposure factors associated with SARS-CoV-2 seroprevalence stratified for univariable and multivariable logistic regression models.**

| | | | | Univariable | | | Multivariable | |
|---|---|---|---|---|---|---|---|---|
| **Gender** | **N** | **SP (%)** | **OR** | **95% CI** | **p value** | **OR** | **95% CI** | **p value** |
| Male (ref.) | 4,167 | 18.7 | - | - | 0.043 | - | - | 0.064 |
| Female | 5,829 | 20.0 | 1.08 | 0.98–01.20 | 0.126 | 1.04 | 0.93–1.16 | 0.490 |
| Other | 5 | - | - | - | - | - | - | - |
| Age (years) | 10,001 | - | 1.00 | 1.00–1.00 | 0.748 | 1.01 | 1.00–1.01 | <0.001 |
| Current education | | | | | | | | |
| No (ref.) | 9,309 | 19.1 | - | - | **0.001** | - | - | **0.008** |
| Other/unknown education | 140 | 17.1 | 0.88 | 0.56–1.36 | 0.555 | 0.92 | 0.58–1.45 | 0.718 |
| Secondary school | 39 | 28.2 | 1.66 | 0.83–3.34 | 0.155 | 1.93 | 0.94–3.96 | 0.074 |
| Secondary vocational education | 66 | 33.3 | 2.11 | 1.26–3.54 | **0.004** | 1.88 | 1.10–3.23 | **0.022** |
| Higher professional education | 232 | 23.3 | 1.28 | 0.94–1.75 | 0.114 | 1.27 | 0.91–1.78 | 0.165 |
| Scientific education | 215 | 26.5 | 1.53 | 1.12–2.07 | **0.007** | 1.60 | 1.14–2.25 | **0.007** |
| Level of education | | | | | | | | |
| Practically trained (ref.) | 4,768 | 20.6 | - | - | - | - | - | - |
| Theoretically trained | 5,233 | 18.4 | 0.87 | 0.79–0.96 | **0.005** | 0.90 | 0.81–1.00 | 0.058 |
| Geography | | | | | | | | |
| Southern part (ref.) | 6,231 | 17.0 | - | - | - | - | - | - |
| Northern part | 3,770 | 23.6 | 1.50 | 1.36–1.66 | **<0.001** | 1.44 | 1.30–1.60 | **<0.001** |
| Work sector | | | | | | | | |
| Other (ref.) | 8,165 | 18.1 | - | - | **<0.001** | - | - | **<0.001** |
| Healthcare other—(partly) worked from home | 500 | 19.0 | 1.06 | 0.84–1.33 | 0.623 | 1.21 | 0.96–1.54 | 0.113 |
| Healthcare other—not worked form home | 364 | 26.6 | 1.64 | 1.29–2.08 | **<0.001** | 1.72 | 1.34–2.20 | **<0.001** |
| Disability care—(partly) worked from home | 83 | 24.1 | 1.43 | 0.86–2.38 | 0.163 | 1.37 | 0.81–2.29 | 0.239 |
| Disability care—not worked from home | 84 | 50.0 | 4.52 | 2.93–6.95 | **<0.001** | 4.17 | 2.68–6.49 | **<0.001** |
| Home care—(partly) worked from home | 41 | 22.0 | 1.27 | 0.61–2.67 | 0.527 | 1.35 | 0.64–2.87 | 0.432 |
| Home care—not worked from home | 98 | 28.6 | 1.81 | 1.16–2.81 | **0.009** | 1.70 | 1.08–2.69 | **0.022** |
| Nursing home care—(partly worked from home) | 61 | 29.5 | 1.89 | 1.09–3.29 | **0.024** | 2.01 | 1.14–3.52 | **0.015** |
| Nursing home care—not worked from home | 102 | 32.4 | 2.16 | 1.42–3.28 | **<0.001** | 2.20 | 1.44–3.38 | **<0.001** |
| Hospital care—(partly) worked from home | 113 | 23.8 | 1.49 | 0.97–2.29 | 0.071 | 1.58 | 1.02–2.46 | **0.041** |
| Hospital care—not worked from home | 121 | 28.1 | 1.77 | 1.18–2.63 | **0.005** | 1.75 | 1.16–2.62 | **0.007** |
| Catering—(partly) worked from home | 182 | 23.6 | 1.40 | 0.99–1.98 | 0.059 | 1.34 | 0.93–1.91 | 0.112 |
| Catering—not worked from home | 87 | 24.1 | 1.44 | 0.88–2.36 | 0.150 | 1.47 | 0.89–2.44 | 0.133 |
| Member association/club | | | | | | | | |
| No member (ref.) | 5,182 | 17.8 | - | - | **<0.001** | - | - | **0.019** |
| Member of one association/club | 3,772 | 20.6 | 1.20 | 1.08–1.34 | **0.001** | 1.09 | 0.98–1.22 | 0.123 |
| Member of two associations/clubs | 925 | 22.6 | 1.35 | 1.14–1.60 | **<0.001** | 1.12 | 0.93–1.34 | 0.240 |
| Member of three/four associations/clubs | 122 | 33.6 | 2.35 | 1.60–3.44 | **<0.001** | 1.82 | 1.22–2.72 | **0.003** |
| Winter sports and travelling abroad | | | | | | | | |
| No winter sports (ref.) | 9,440 | 19.0 | - | - | **<0.001** | - | - | **<0.001** |
| No après-ski during winter sports Austria | 163 | 20.2 | 1.08 | 0.73–1.59 | 0.697 | 1.22 | 0.83–1.81 | 0.313 |
| A few days après-ski during winter sports Austria | 158 | 22.2 | 1.21 | 0.83–1.77 | 0.324 | 1.26 | 0.85–1.86 | 0.246 |

*(Continued)*

**Table 2.** (Continued)

| Gender | N | SP (%) | OR | Univariable | | OR | Multivariable | |
|---|---|---|---|---|---|---|---|---|
| | | | | 95% CI | p value | | 95% CI | p value |
| Majority of the days après-ski during winter sports Austria | 240 | 34.6 | 2.25 | 1.72–2.95 | **<0.001** | 2.49 | 1.88–3.29 | **<0.001** |
| Travelled abroad to Spain | 282 | 24.1 | 1.33 | 1.00–1.75 | **0.047** | 1.41 | 1.06–1.87 | **0.019** |
| Celebrating carnival | | | | | | | | |
| Did not celebrate carnival (ref.) | 4,725 | 17.9 | - | - | **<0.001** | - | - | 0.088 |
| Celebrated less than or 3 hours inside | 1,373 | 19.0 | 1.07 | 0.92–1.25 | 0.369 | 1.09 | 0.93–1.27 | 0.310 |
| Celebrated between 3 and 8 hours inside | 1,288 | 19.4 | 1.10 | 0.94–1.29 | 0.228 | 1.08 | 0.92–1.28 | 0.329 |
| Celebrated between 8 and 18 hours inside | 1,422 | 21.7 | 1.27 | 1.10–1.47 | **0.001** | 1.19 | 1.02–1.40 | **0.026** |
| Celebrated more than 18 hours inside | 1,193 | 23.5 | 1.40 | 1.20–1.63 | **<0.001** | 1.24 | 1.04–1.47 | **0.016** |
| Specific activities February-March 2020 | | | | | | | | |
| Singing activities[a] | 389 | 28.8 | 1.71 | 1.37–2.15 | **<0.001** | 1.65 | 1.30–2.08 | **<0.001** |
| Attended funeral | 504 | 23.2 | 1.27 | 1.02–1.57 | **0.030** | 1.08 | 0.86–1.35 | 0.494 |
| Played wind instrument in interplay | 164 | 27.4 | 1.58 | 1.12–2.23 | **0.010** | 1.26 | 0.88–1.81 | 0.210 |
| Visited sport event (except soccer game) | 323 | 26.0 | 1.47 | 1.14–1.90 | **0.003** | 1.18 | 0.90–1.55 | 0.234 |
| Practiced gymnastics | 110 | 27.3 | 1.56 | 1.02–2.38 | **0.039** | 1.22 | 0.78–1.90 | 0.379 |
| Practiced ball sports (except soccer) | 174 | 36.2 | 2.39 | 1.75–3.27 | **<0.001** | 1.90 | 1.36–2.65 | **<0.001** |
| Visited a bar or café | 1,657 | 23.4 | 1.33 | 1.17–1.51 | **<0.001** | 1.13 | 0.96–1.32 | 0.145 |
| Visited a club or disco | 294 | 26.2 | 1.49 | 1.14–1.94 | **0.003** | 1.23 | 0.90–1.66 | 0.189 |
| Went out for dinner | 2,294 | 22.0 | 1.23 | 1.09–1.37 | **<0.001** | 1.03 | 0.89–1.19 | 0.691 |
| Took a day trip with bus or boat | 50 | 34.0 | 2.14 | 1.19–3.85 | **0.011** | 1.78 | 0.97–3.27 | 0.064 |
| Constant | | | | | | 0.09 | | 0.000 |

*P-values<0.05 are shown in bold.

Note.- CI, confidence intervals; OR, odds ratio; Ref., reference category; SARS-CoV-2, Severe Acute Respiratory Syndrome Coronavirus-2; SP, seroprevalence.

[a] Being member of a choral society, (church)choir or participating in singing activity in February-March 2020.

Multivariable logistic regression model is additionally corrected for age, gender, level of education and geographical region. Overall Cox $R^2$ of the multivariate model was 0.032.

The odds of seropositivity was 1.55 [95% CI:1.37–1.76] for participants working in healthcare versus participants not working in healthcare. The odds were notably higher in disability care (OR = 4.17 [95% CI:2.68–6.49]). Overall, not working from home was associated with an increased odds in all healthcare sectors.

Participants being a member of three or four associations or clubs (listed in methods section) were more likely to have COVID-19 antibodies (OR = 1.82 [95% CI:1.22–2.72]. Singing activities also increased odds of seropositivity (OR = 1.65 [95% CI:1.30–2.08]).

**Specific factors for February-March 2020.** Visiting an après-ski bar for the majority of the days during winter sports in Austria was associated with increased odds for seropositivity (OR = 2.49 [95% CI:1.88–3.29]). In addition, traveling to Spain was established to be associated (OR = 1.41 [95% CI:1.06, 1.87]).

Carnival celebrations while spending more than eight hours inside was positively associated with seropositivity ($OR_{between\ 8\ and\ 18\ hours}$ = 1.19 [95% CI:1.02–1.40] and $OR_{more\ than\ 18\ hours}$ = 1.24 [95% CI:1.04–1.47].

Of the various other activities evaluated, the following increased odds for seropositivity: attended a funeral, played a wind instrument in interplay, visited a sport event (except soccer game), practiced gymnastics, practiced a ball sport (except soccer), visited a bar, café, club, or disco, went out for dinner, and took a day trip by bus or boat.

Practicing a ball sport (except soccer) was associated with seropositivity in the multivariate model (OR = 1.90 [95% CI:1.36–2.65]).

**Notable activities with insufficient participants.**   An exceptionally high seroprevalence appeared among the participants who attended a charity event in Kessel, a small village in the middle of the study province, which was more than four times the average seroprevalence. Only 15 participants attended this event, making this result sensitive to bias. Two other specific events, namely visiting professional soccer matches in Maastricht, resulted in seroprevalences up to twice the average. Likewise, the number of participants was too small for these events.

## Discussion

In this cross-sectional study evaluating the seroprevalence and extensive questionnaire data from 10,000 inhabitants of a southern Dutch province, the seroprevalence was almost 20% by the end of 2020. Several exposure factors were independently associated with seropositivity: following secondary vocational or scientific education, working in healthcare and not working from home, and being a member of three or four associations or clubs. Specifically, for the months of February-March 2020, relevant exposure factors included visiting an après-ski bar in Austria for the majority of the days during winter sports, travelling to Spain, celebrating carnival for longer than eight hours inside, participating in a singing activity, and practicing a ball sport. The majority of the independently associated exposure factors established in our study reflect circumstances where social distancing is probably not generally maintained. Moreover, circumstances where participants are thought to have a high contact rate and activities performed inside show clear associations with seropositivity.

The fact that the initial strict Dutch testing policy greatly underestimated cumulative infections, is highlighted by the results of our study. Until 1st December 2020, in total 25,592 PCR confirmed COVID-19 cases were reported among 1.12 million inhabitants of the study province, reflecting 2.3% of all inhabitants [3]. The seroprevalence calculated in our study is almost ten times higher compared to the PCR confirmed COVID-19 cases since 27th February 2020 in the study province (2.3% versus 19.5%).

Other Dutch seroprevalence studies established seroprevalences between 11% and 12% in healthy plasma donors in November 2020, and between 5% and 10% in a representative sample of Dutch inhabitants in September 2020 [15, 16]. Overall, our study established the highest seroprevalence in the study province. However, it should be considered that due to the convenience sample, there is a possibility that people who expected to have had an COVID-19 infection were more likely to participate. This could result in a slight overestimation of the seroprevalence in our study.

The effect of not maintaining social distancing can be seen in occupational settings. Working in healthcare often requires direct patient contact which makes social distancing impossible. Several studies have identified an increased risk of COVID-19 infection in healthcare [9, 10, 16]. The possibility of encountering an infected patient is high among healthcare workers, as severely ill patients are admitted to healthcare facilities to receive required treatment [17].

In addition to the nature of a specific occupation itself, the possibility of working from home diminishes occupational exposure risk. A large Dutch cross-sectional population-based study supports this, by showing 0.71 times lower odds of seropositivity among participants working from home compared to participants not working from home, independent of work sector. The reason for the decreased risk may be partly the result of reduced daily physical contact with colleagues or clients. Seroprevalence was namely also 0.61 times lower among participants without physical contact with patients or clients in their professional or voluntary work [16].

Social distancing is closely related to the contact rate. A reduction of 71% in the average number of community contacts was observed, since social distancing measures were implemented in the Netherlands [18]. Circumstances where participants were thought to have a great contact rate showed a higher seroprevalence in our study, including being a member of three or four clubs or associations. Attending social meetings equals more varying social contacts, in turn increasing the possibility of getting in contact with a COVID-19 infected person. This is supported by a previous study examining the association between network parameters and several self-reported infections [19]. Among 3,004 Dutch participants aged 60 years on average, network size was significantly associated with upper respiratory tract and gastrointestinal infection. For every additional 10% of acquaintance contacts, including club mates, the odds of lower respiratory tract infection increased by 4%. However, this study did not show a significant association between number of club memberships and any of the above-mentioned infections.

The probability of virus transmission in a contact is partly determined by the stability of the virus in the environment, which is greater in circumstances with insufficient ventilation. Spending time inside, for example celebrating carnival or going to an après-ski bar during winter sports, shows a clear association with seropositivity in our study. One comparable study regarding spending time inside was carried out among 1,120 Danish medical students. Students who attended one or two parties organized before lockdown happened demonstrated a sixfold increase of seropositivity [20]. These findings highlight that an indoor environment is favourable for SARS-CoV-2 transmission. Nevertheless, no significant association was established between visiting a regular bar, café, or disco and seropositivity in our study. This might be explained by the large proportion of participants that undertook this type of activity, leading to an approximate even distribution between seropositive and seronegative participants. As this activity is not very specific, the association might be faded by other more pronounced activities and circumstances. Moreover, the counted visits to a bar, café, or disco not related to the carnival celebrations were in all probability before COVID-19 was widely spread in the study province. Therefore, there is a lower chance of getting infected during this kind of visit. Subsequently, implementation of infection prevention measures totally prohibited visiting these facilities, thereby eliminating exposure risk in this setting.

Furthermore, the impact of effortless travel nowadays should not be underestimated. According to an American mathematical modelling study, unconstrained mobility would have significantly accelerated the spread of SARS-CoV-2, especially in Central Europe, Spain and France [21]. The significant associations with travel destinations determined in our study support this.

The possible indirect impact of the COVID-19 pandemic on general health and lifestyle, for instance provoked by lockdowns, implementation of infection prevention measures and required self-isolation, has to be addressed. Increased alcohol consumption or intake of illegal substances–known to have a detrimental effect on health–have been shown during the second emergency state of the pandemic [22]. In contrast, the indirect impact of the pandemic could be effectively counteracted by stimulating the intake of a balanced diet, resulting in symbiosis by modulating the gut microbiota [23]. It is crucial to find the right balance between combatting waves of infection and thereby protecting public health on one side, and limiting the detrimental indirect effect of lockdowns and infection prevention measures on health on the other side.

To the best of our knowledge, this is the first large scale study examining an extensive diversity of possible demographic, social and behavioural risk factors for SARS-CoV-2 seropositivity. The questionnaire mapped many exposure factors and gave rise to the possibility to correct analyses for relevant characteristics. Regarding the validity of the serological test used in our study, a nationwide multicentre evaluation study indicated high sensitivity (97.5% after severe

infection and 95.4% after mild infection) and high specificity of 99.6%, using PCR as reference [13]. Solely serology could be used to estimate cumulative infections, as initially a strict testing policy was maintained in the Netherlands. All tests were performed in the same laboratory, limiting differences in processing and execution of the tests, making results well comparable.

Our study had some limitations as well. First, selection bias cannot be ruled out due to the convenience sampling. However, sampling methods including a random sample can be subjected to substantial non-response, resulting in bias as well. The yielded study population in our study is quite representative regarding geographical and age distribution. Considering we were interested in exposure factors, the generalizability of the determined seroprevalence was less relevant. Second, recall bias could have occurred since participating in specific activities was listed for a period up to eight months earlier. Associations can be underestimated when participants did not remember their attendance at a specific event. In the study design, we attempted to limit recall bias by encouraging participants to use their agenda when filling out the questionnaire. Completing the questionnaire at home diminished time pressure and lowered possible recall bias as well. Third, associations between exposure factors at the beginning of the pandemic and seropositivity might be attenuated, due to the possible long period between the exposure and serology testing. Related to this, the overall multivariate model has a low $R^2$ of 3.2%, meaning that a small part the variance in the data can be explained by the final model. This indicates that there are many more factors that contributed. From source and contact tracing activities we know that many infections occur, for instance, at home or during contacts with friends. Factors like cohabitation and the presence of children that go to school, but also general health before COVID-19, may predispose individuals to COVID-19 infection risk. In an eight-month period, there are innumerable activities and circumstances where an infection can be acquired. Nevertheless, exposure factors that took place before the official first COVID-19 cases, for example the carnival celebrations, were still independently associated, implying that these factors have substantially contributed to the primary spread of the virus.

In conclusion, the strict Dutch testing policy resulted in a great underestimation of cumulative COVID-19 infections during the first wave. Our results confirm that relevant COVID-19 exposure factors generally reflect circumstances where social distancing was impossible, and the number and duration of contacts was high, in particular for indoor activities (without proper ventilation). The measures taken at the beginning of the pandemic accurately targeted these circumstances to contain virus transmission. Confirmation of these prevention measures is of great value to take lessons from the initial response to the pandemic, and thereby provides guidance on steps to take in new waves of infection with new virus variants.

Moreover, our results can have an added value in responsibly relaxing infection prevention measures and reopening society, as they can help to prioritize which activities should be addressed as high COVID-19 risk. Subsequently, the risk of exponential spreading can be managed when allowing these 'high risk' activities again.

## Acknowledgments

We gratefully acknowledge CJD Goense and LCJ Steijvers for their valuable contribution to the development of the questionnaire, and the technicians of the Department of Medical Microbiology subdivision serology of the MUMC+.

## Author Contributions

**Conceptualization:** Demi M. E. Pagen, Stephanie Brinkhues, Nicole H. T. M. Dukers-Muijrers, Casper D. J. den Heijer, Noortje Bouwmeester-Vincken, Daniëlle A. T. Hanssen, Inge H. M. van Loo, Paul H. M. Savelkoul, Christian J. P. A. Hoebe.

**Formal analysis:** Demi M. E. Pagen, Stephanie Brinkhues, Nicole H. T. M. Dukers-Muijrers, Casper D. J. den Heijer, Christian J. P. A. Hoebe.

**Funding acquisition:** Christian J. P. A. Hoebe.

**Investigation:** Demi M. E. Pagen, Stephanie Brinkhues, Daniëlle A. T. Hanssen.

**Methodology:** Demi M. E. Pagen, Stephanie Brinkhues, Nicole H. T. M. Dukers-Muijrers, Noortje Bouwmeester-Vincken, Daniëlle A. T. Hanssen, Inge H. M. van Loo, Christian J. P. A. Hoebe.

**Project administration:** Demi M. E. Pagen, Stephanie Brinkhues.

**Resources:** Demi M. E. Pagen.

**Supervision:** Stephanie Brinkhues, Nicole H. T. M. Dukers-Muijrers, Casper D. J. den Heijer, Paul H. M. Savelkoul, Christian J. P. A. Hoebe.

**Visualization:** Demi M. E. Pagen.

**Writing – original draft:** Demi M. E. Pagen.

**Writing – review & editing:** Demi M. E. Pagen, Stephanie Brinkhues, Nicole H. T. M. Dukers-Muijrers, Casper D. J. den Heijer, Noortje Bouwmeester-Vincken, Daniëlle A. T. Hanssen, Inge H. M. van Loo, Paul H. M. Savelkoul, Christian J. P. A. Hoebe.

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
