## [Decision Letter · Decision Letter 0]

25 Mar 2022

PONE-D-22-02052Exposure factors associated with SARS-CoV-2 seroprevalence during the first eight months of the COVID-19 pandemic in the Netherlands: a cross-sectional studyPLOS ONE

Dear Dr. Pagen,

Thank you for submitting your manuscript to PLOS ONE. After careful consideration, we feel that it has merit but does not fully meet PLOS ONE’s publication criteria as it currently stands. Therefore, we invite you to submit a revised version of the manuscript that addresses the points raised below during the review process. Please submit your revised manuscript by May 09 2022 11:59PM. If you will need more time than this to complete your revisions, please reply to this message or contact the journal office at plosone@plos.org. Please include the following items when submitting your revised manuscript:A rebuttal letter that responds to each point raised by the academic editor and reviewer(s). You should upload this letter as a separate file labeled 'Response to Reviewers'.A marked-up copy of your manuscript that highlights changes made to the original version. You should upload this as a separate file labeled 'Revised Manuscript with Track Changes'.An unmarked version of your revised paper without tracked changes. You should upload this as a separate file labeled 'Manuscript'.

We look forward to receiving your revised manuscript.

Kind regards,

Ray Borrow, Ph.D., FRCPath

Academic Editor

PLOS ONE

Journal Requirements:

Reviewers' comments:

Reviewer's Responses to Questions

**Comments to the Author**

1. Is the manuscript technically sound, and do the data support the conclusions?

Reviewer #1: Yes

Reviewer #2: Yes

2. Has the statistical analysis been performed appropriately and rigorously? 

Reviewer #1: Yes

Reviewer #2: Yes

3. Have the authors made all data underlying the findings in their manuscript fully available?

Reviewer #1: Yes

Reviewer #2: Yes

4. Is the manuscript presented in an intelligible fashion and written in standard English?

Reviewer #1: Yes

Reviewer #2: Yes

5. Review Comments to the Author

Reviewer #1: Dear author,

The paper entitled “Exposure factors associated with SARS-CoV-2 seroprevalence during the first eight months of the COVID-19 pandemic in the Netherlands: a cross-sectional study” is interesting despite some limitations. This paper should be review accord with comments below.

Abstract

The background of this paper should be rewritten to indicate the main aim of this research, because is not clear the objective of this research.

In the methods is refer a sample with 10,000 inhabitants, but the results are reported to the 10,001, is confused. The authors should to explain in the methods all inhabitants that was used to this study.

Introduction

The introduction is clear and is appropriated to this research, however may be mentioned the implications in the population provoked by COVID-19 infection, such as, lockdown in the home. Include the reference doi: 10.1371/journal.pone.0260322. in the paper.

Methods

In the online questionnaire was ask “who do you live with their”? And, if have children that go to school?

The blood samples were used only to SARS-CoV-2-ab ELISA test? Which amount of blood sample was need to proceed the ELISA test? Were done duplicates?

The samples that obtain results in the borderline, were test again? Or used other kit or technique? Why you don´t confirmed these results with the PCR assay?

Results

In the figure 2 can you explain who are reserve list participants? The profile of these participants is similar to the other participants (10,000 initial)?

Which the variables do you used to multivariate analysis?

Discussion

The online questionnaire asked about the health condition of participants? The health condition (before COVID-19) the participants is important because may be predispose the individuals to COVID-19 infection – this theme should be addressed in the discussion. The same to the cohabitation and the presence of children that go to the school.

Reviewer #2: In this paper entitled "Exposure factors associated with SARS-CoV-2 seroprevalence during the first eight months of the COVID-19 pandemic in the Netherlands," the authors investigated a wide variety of demographic, behavioral, and social exposure factors associated with seropositivity. The results indicated that 19.5 % of the participants were seropositive. The manuscript has potential but has some issues that have to be addressed for the manuscript to be published in PLOS ONE journal.

Minor comments:

1) Did the author ask about the vaccination status of the participants?

2) Introduction Section: the authors should provide information like mortality rate, incubation, various initial prevention approaches, and diet and natural biomolecules for improving immunity and health (doi: 10.1007/s12088-020-00908-0).

3) Introduction Section: Minor information on the variants of COVID-19 and their future challenges can be included i.e. doi: 10.1007/s15010-021-01734-2.

4) The authors should cross-check all abbreviations in the manuscript. Initially, define in the full name followed by abbreviation.

5) Highlight the importance of the study in the manuscript.

6. PLOS authors have the option to publish the peer review history of their article (what does this mean?). If published, this will include your full peer review and any attached files.

Reviewer #1: No

Reviewer #2: **Yes: **Aditya Kumar Sharma

---

## [Author Response · Author response to Decision Letter 0]

7 Apr 2022

Dear Dr. Borrow, 

Thank you for giving us the opportunity to revise our manuscript. 

We also thank the reviewers for their helpful comments, which we have addressed in the revised manuscript and in our point-by-point reply below. 

We hope that our comments are satisfactory and that the paper is now acceptable for publication in PLOS ONE. 

Yours sincerely,

On behalf of all coauthors,

Demi Pagen 

Response to the Editor:

Thank you for indicating inaccuracies according to the style requirements. All style requirements are checked and adjusted according to the templates. 

Thank you for pointing out the unacceptability of our data availability statement. There are legal restrictions on sharing data publicly, as the data of our study contain potentially identifying and sensitive participant information. Due to the General Data Protection Regulation, it is not allowed to distribute or share any personal data that can – directly or indirectly – be traced back to an individual. Besides, publicly sharing the data would not be in accordance with participants’ consent obtained for this study. Therefore, data are available from the head of the data-archiving of the Public Health Service South Limburg on reasonable request (Helen Sijstermans: helen.sijstermans@ggdzl.nl). This statement is in line with previously published articles of our research group (https://doi.org/10.1371/journal.pone.0258701). 

Thank you for highlighting incorrect references included in the reference list. All references are reviewed and replaced or adjusted according to the requirements. Adjusted references in the Revised Manuscript include reference 7 (page 21 line 457), 9 (page 22 line 460-462), 10 (page 22 line 463-464), 11 (page 22 line 465-468), 16 (page 23 line 482-485), and 20 (page 23 line 494-496).

Response to reviewers:

Reviewer #1: Dear author,

The paper entitled “Exposure factors associated with SARS-CoV-2 seroprevalence during the first eight months of the COVID-19 pandemic in the Netherlands: a cross-sectional study” is interesting despite some limitations. This paper should be review accord with comments below.

Thank you for the complements. We reviewed the paper according to the comments and incorporated them into the Revised Manuscript.

Abstract

1. The background of this paper should be rewritten to indicate the main aim of this research, because is not clear the objective of this research. In the methods is refer a sample with 10,000 inhabitants, but the results are reported to the 10,001, is confused. The authors should to explain in the methods all inhabitants that was used to this study.

Thank you for pointing out indistinctness about the study aim and obtained study population. We elaborated on both points – by stating the research aim more pronounced and explaining the use of a reserve list – to enhance clarity in the Revised Manuscript; Abstract section Background (page 2 line 37-41) and section Method (page 2 line 46-49). 

Introduction

2. The introduction is clear and is appropriated to this research, however may be mentioned the implications in the population provoked by COVID-19 infection, such as, lockdown in the home. Include the reference doi: 10.1371/journal.pone.0260322. in the paper.

Thank you for this suggestion, we agree that the indirect impact of the COVID-19 pandemic on general health and lifestyle cannot be denied. Therefore, we added a statement acknowledging this point including the reference in the Revised Manuscript; Discussion section (page 19 line 378-382).

Methods

3. In the online questionnaire was ask “who do you live with their”? And, if have children that go to school?

Thank you for pointing this out. In the questionnaire, we determined participants home situation by asking whether they were married/living together or single/widow. Besides, we questioned whether the participants had children (living at home/not living at home) or not. It was not further specified if the children were currently going to school.

We combined the information about their marital status and having children into one variable to get a better picture of the participant’s home situation and determined seroprevalence per situation (Table 1).

Table 1. Seroprevalence by home situation

Home situation n Seroprevalence (%)

Single/widow – no children 1,291 20.6

Single/widow – children not living at home 399 19.3

Single/widow – children living at home 336 11.6

Married/living together – no children 1,483 19.4

Married/living together – children not living at home 2,679 20.3

Married/living together – children living at home 3,813 19.3

Being single/widow and having children living at home, resulted in a significantly lower seroprevalence. However, as we were interested in investigating factors that resulted in a higher risk of being exposed – reflected by a higher seroprevalence compared to the average of 19.5% in the overall population - we did not take home situation into account in further analysis, due to the fact that it did not contribute to a higher exposure risk.

4. The blood samples were used only to SARS-CoV-2-ab ELISA test? Which amount of blood sample was need to proceed the ELISA test? Were done duplicates?

The blood samples (one 10 ml EDTA tube per participant) were only collected to perform the SARS-CoV-2 Ab ELISA antibody test. This test was used as it showed the best performance in a multicenter evaluation by the National Institute for Health and Environment. A minimum of 100 μl serum per well was needed to perform the Wantai ELISA. Each sample was tested once, as our validation showed the variance of duplicates within one run to be between 1.7-2.4%. Therefore, we determined that testing each sample once was sufficient. 

5. The samples that obtain results in the borderline, were test again? Or used other kit or technique? Why you don´t confirmed these results with the PCR assay?

Thank you for this suggestion. Borderline values were not tested again. Based on pilot experience using the Wantai among five hospital employee with serum pairs, four out of five became positive. Therefore, we decided to classify the borderline values as positives. Due to the small number of borderline values, an overestimation of the seroprevalence seemed negligible.

PCR testing probably would not have resulted in an exclusive answer, as PCR testing is only capable of indicating a current infection, and participants could have been infected up to eight months before the blood drawing and antibody testing was performed. It is acceptable to assume that participants with borderline results had been infected several months before, probably reflecting declining antibody levels, as sero-reconversion occurs over time. Due to this, we did not consider PCR testing as an alternative to indicate whether participants with borderline values had (previously) been infected. 

Results

6. In the figure 2 can you explain who are reserve list participants? The profile of these participants is similar to the other participants (10,000 initial)?

Thank you for this suggestion, we appreciate that you indicate unclarity about the inclusion of the reserve participants. The first 10,000 registrations were determined to be the initial registrations. The registration link automatically allocated registrations beyond the first 10,000 to the reserve list (maximum capacity of 3,000 registrations). In order to assure sufficient participants (minimally 10,000) with complete participation (both questionnaire and blood drawing; antibody testing), reserve participants were invited to participate when initial registered participants declined participation. The resevres were invited on first-come-first-serve base, so no selection was made according to demographic factors. This is clarified in the Revised Manuscript; Abstract section Method (page 2 line 46-49) and Method section subheading Participants (page 5 line 134-137).

Despite we did not take demographic factors into account when inviting reserve participants, here we describe the demographic factors of the initial and reserve list participants, showing a comparable age and sex distribution between initial and reserve list registrations (Table 2).

Table 2. Age and sex distribution of initial, reserve list and overall participants

 Initial (n=9,252) Reserve list (n=749) Participants (n=10,001)

Sex, n(%) 

 Women 5,380 (58.1) 449 (59.9) 5,829 (58.3)

 Men 3,870 (41.8) 297 (39.7) 4,167 (41.7)

 Other 2 (0.1) 3 (0.4) 5 (0)

Age, mean (SD) 50 (15) 48 (15) 49 (15)

Age, n(%) 

 18-29 1,106 (12.0) 102 (13.6) 1,208 (12.1)

 30-39 1,547 (16.7) 129 (17.2) 1,676 (16.8)

 40-49 1,626 (17.6) 140 (18.7) 1,766 (17.7)

 50-59 2,218 (24.0) 175 (23.4) 2,393 (23.9)

 60-69 2,018 (21.8) 145 (19.4) 2,163 (21.6)

 70-79 695 (7.5) 51 (6.8) 746 (7.5)

 80+ 42 (0.5) 7 (0.9) 49 (0.5)

7. Which the variables do you used to multivariate analysis?

All investigated exposure factors statistical significantly (p<0.05) associated with seropositivity in univariate analyses (indicating a higher risk of exposure) and general confounders (age and sex) were maintained in the multivariate model. These included: sex, age, level of education, geographical region, working from home per work sector, currently following education, being member of multiple associations/clubs, travelled for winter sports to Austria and visited an après-ski bar, travelled to Spain, celebrated carnival, attended or participated in singing activities, attended a funeral, played a wind instrument in interplay, visited a sport event, practiced gymnastics, practiced a balls sport, visited a bar or café, visited a club or disco, went out for dinner, and took a day trip with bus or boat. The selection procedure of variables to be maintained in the multivariable model is described in the Manuscript; Method section subheading Statistical analysis (page 8 line 205-209).

Discussion

8. The online questionnaire asked about the health condition of participants? The health condition (before COVID-19) the participants is important because may be predispose the individuals to COVID-19 infection – this theme should be addressed in the discussion. The same to the cohabitation and the presence of children that go to the school.

Thank you for these suggestions. We recognize the importance about general health in relation to COVID-19 predisposition. The questionnaire included one question on the subjective assessment of the participants general health: ‘’In general, I think my health is: excellent, very good, good, moderate, or bad’’. However, as the exact moment of infection was generally not known, it cannot be excluded that the general health status was already affected by the previous infection in our study. Therefore, correcting for general health before the infection was challenging. However, while keeping the previously mentioned limitation into account, we investigated whether serostatus differed according to general health status, which resulted in no significant differences (Chi-square=6.458 p=0.167; excellent n=1,286 SP=20.6%, very good n=3,251 SP=20.3%, good n=4,517 SP=19.1%, moderate n=878 SP=17.0% and bad n=69 SP=17.4%). Therefore, general health score was not included in further analysis. Furthermore, general health before the COVID-19 pandemic was not asked, so unfortunately we were not able to include this in our analysis. 

We included a statement on the importance of general health and home situation (having children going to school) in the Revised Manuscript; Discussion section (page 20 line 416-418). 

 

Reviewer #2: In this paper entitled "Exposure factors associated with SARS-CoV-2 seroprevalence during the first eight months of the COVID-19 pandemic in the Netherlands," the authors investigated a wide variety of demographic, behavioral, and social exposure factors associated with seropositivity. The results indicated that 19.5 % of the participants were seropositive. The manuscript has potential but has some issues that have to be addressed for the manuscript to be published in PLOS ONE journal.

Thank you for your compliments. We addressed the issues you pointed out in our Revised Manuscript.

Minor comments:

1) Did the author ask about the vaccination status of the participants?

Thank you for this suggestion. The study was conducted at the end of 2020 (November – December), before vaccination started in the Netherlands. Therefore, the questionnaire did not include any questions on previous COVID-19 vaccination, as this was not relevant at that time.

2) Introduction Section: the authors should provide information like mortality rate, incubation, various initial prevention approaches, and diet and natural biomolecules for improving immunity and health (doi: 10.1007/s12088-020-00908-0).

Thank for your suggestion to include information on diet in relation to improving immunity and health. We included and statement about the importance of a balanced diet in relation to health and the reference in the Revised Manuscript; Discussion section (page 19 line 382-384).

3) Introduction Section: Minor information on the variants of COVID-19 and their future challenges can be included i.e. doi: 10.1007/s15010-021-01734-2.

Thank you for your suggestion to incorporate information about virus variants. We included this and the reference in the Revised Manuscript; Introduction section (page 3 line 77-80).

4) The authors should cross-check all abbreviations in the manuscript. Initially, define in the full name followed by abbreviation.

Thank you for indicating overlooked errors in the use of abbreviations. We corrected the following abbreviations in the Revised Manuscript; 

Introduction section (page 3 line 69):

- coronavirus disease (COVID-19) adjusted into coronavirus disease 2019 (COVID-19)

Method section subheading Data collection (page 6 line 148-149 and 152-153): 

- EDTA adjusted into ethylenediaminetetraacetic acid (EDTA)

- ELISA adjusted into enzyme-linked immunoassay (ELISA)

Method section subheading Statistical analysis (page 9 line 213):

- SPSS adjusted into Statistical Package for the Social Sciences (SPSS)

Results section subheading Seroprevalence (page 12 line 247):

- CI adjusted into Confidence Interval (CI)

Results section subheading Exposure factors associated with seropositivity (page 12 line 257):

- OR adjusted into Odds Ratio (OR)

5) Highlight the importance of the study in the manuscript.

Thank you for your suggestion to highlight the importance of our study. We emphasized the importance multiple times in the Revised Manuscript; Introduction section (page 4 line 78-80 and 5 line 111-114) Discussion section (page 20 line 429-431).

---

## [Editor Report · Decision Letter 1]

22 Apr 2022

Exposure factors associated with SARS-CoV-2 seroprevalence during the first eight months of the COVID-19 pandemic in the Netherlands: a cross-sectional study

PONE-D-22-02052R1

Dear Dr. Pagen,

We’re pleased to inform you that your manuscript has been judged scientifically suitable for publication and will be formally accepted for publication once it meets all outstanding technical requirements.

Kind regards,

Ray Borrow, Ph.D., FRCPath

Academic Editor

PLOS ONE
---

## [Editor Report · Acceptance letter]

4 May 2022

PONE-D-22-02052R1 

Exposure factors associated with SARS-CoV-2 seroprevalence during the first eight months of the COVID-19 pandemic in the Netherlands: a cross-sectional study 

Dear Dr. Pagen:

I'm pleased to inform you that your manuscript has been deemed suitable for publication in PLOS ONE. Congratulations! Your manuscript is now with our production department. 

Kind regards, 

on behalf of

Prof. Ray Borrow 

Academic Editor

PLOS ONE